# Team Workload and Performance of Healthcare Workers with Musculoskeletal Symptoms

**DOI:** 10.3390/ijerph20010742

**Published:** 2022-12-31

**Authors:** Elamara Marama de Araújo Vieira, Jonhatan Magno Norte da Silva, Wilza Karla dos Santos Leite, Ruan Eduardo Carneiro Lucas, Luiz Bueno da Silva

**Affiliations:** 1Physiotherapy Department, Universidade Federal da Paraíba, João Pessoa 58051-900, Brazil; 2Campus do Sertão, Universidade Federal de Alagoas, Delmiro Gouveia 57480-000, Brazil; 3Department of Biological and Health Sciences, Universidade Federal do Amapá, Macapá 68903-419, Brazil; 4Campus Parauapebas, Universidade Federal Rural da Amazônia, Parauapebas 68515-000, Brazil; 5Production Engineering Department, Universidade Federal da Paraíba, João Pessoa 58051-900, Brazil

**Keywords:** occupational health, workplace, musculoskeletal disorders, pain, ergonomics, healthcare providers

## Abstract

In healthcare professionals, musculoskeletal complaints are the most frequent health disorders with the greatest potential for productivity losses. The teamwork developed by these professionals can be a coping strategy, but it can also be one more demand for the maintenance of performance. For this reason, this research aimed to investigate the relationship between team workload and performance in healthcare workers with different intensities of musculoskeletal symptoms. A survey was conducted with health professionals from 24 institutions of the Brazilian public health system, recruited by stratified probability sampling. Through non-hierarchical cluster analysis, the sample was allocated into three groups based on the intensity of musculoskeletal symptoms. We analyzed the approximation between the variables of “team workload” and “performance” of the groups formed in the previous phase through multiple correspondence analysis. In the group with higher musculoskeletal symptom scores, there was lower performance and a worse team workload. As the intensity of symptoms decreased, team workload and performance became closer variables in a two-dimensional space, indicating that the relationship between team workload and performance is improved in situations of low musculoskeletal symptom intensity.

## 1. Introduction

Workers’ health conditions have become an increasingly popular human product indicator. Their repercussions in the world of work have aroused the interest of various stakeholders when it comes to the construction of strategies to minimize damage, considering that having sick workers can bring socioeconomic consequences, with losses for them, for the health care services, and society [1].

In health services, the loss of productivity is a critical factor because it reflects on patient safety and the quality of care provided. Several studies have registered multi-symptomatic health professionals, with complaints especially associated with fatigue and the ability to meet the physical demands of the job, affecting the preservation of the worker’s ability to perform the task [1,2,3,4,5].

Therefore, it is understood that the health of health professionals and the quality of their work are interconnected variables, which involve, on the worker’s part, having good physical and functional capacity. However, in addition to the impact of the disease itself, these abilities can be harmed by work overload, the excessive cognitive and emotional complexity of the task, lack of social support in teamwork, organizational instability, mistrust, and frustration [6].

Having sick professionals on the team requires collective self-regulation as a coping strategy to maintain productivity [7,8]; however, the sharing of tasks in these situations generates an additional work demand by affecting the relationships of the team members and those with the work tools [9]. Thus, we have a workload that is no longer focused on the task at hand, but on the team itself or arising from team relationships, and that, according to Cui et al. [10], can influence the team’s performance in the execution of responsibilities in complex scenarios.

Whereas the “implementation of a culture of healthy work environments that can improve worker performance, productivity and well-being” is one of the current priorities in occupational health indicated by the 6th International Congress on Work Organization and Psychosocial Factors held in 2017 [11], efforts should then be directed at developing a better understanding of the transformations in the work context in adverse situations, such as those caused by illness. 

In healthcare professionals, musculoskeletal complaints are the most frequent health disorders and with the greatest potential for production losses [2]. They are the most frequent causes of presenteeism in some developed countries [12], being significantly associated with work demand, task allocation, and teamwork environment [13], and can be used as an indicator of physical health [14].

In macroeconomic terms, in Brazil between 2012 and 2016, social costs related to musculoskeletal disorders amounted to $2.2 billion, and productivity losses arising from this type of disorder accounted for 79% of costs, with absenteeism at 59 million days [15]. In Chile, the expected annual cost due to chronic musculoskeletal pain was estimated to be $1387.2 million, equivalent to 0.417% of the national GDP (Gross Domestic Product). Of all the causes evaluated, productivity losses due to musculoskeletal disorders were the most significant [12].

For these reasons, we propose that the approximation between team workload and performance in these professionals may be disadvantaged by the severity of musculoskeletal symptom presentation. This proposition is composed of the following hypotheses (H):

**Hypothesis 1.** 
*The degree of severity of musculoskeletal symptoms can be obtained through a score that combines the presence and intensity of these symptoms in different body parts.*


**Hypothesis 2.** 
*The musculoskeletal symptoms severity score distinguishes groups of workers.*


**Hypothesis 3.** 
*Team workload and performance may be disadvantaged in groups of workers with high scores of musculoskeletal symptoms.*


Although there is current research that focuses on analyzing the performance of workers with physical symptoms, such as Van Tilburg et al. [16], Ishimaru et al. [17], and Cochrane et al. [18], none to our knowledge has considered the relationship between team workload and performance in a symptomatic background. Considering that working even in the presence of major health symptoms has been shown to be a social behavioral norm in the healthcare workforce, linked to the thought of responsible behavior toward the team and toward the patient [19,20], it is essential to analyze its moderating role under the various work contexts.

Therefore, our goal was to investigate the relationship between team workload and performance in healthcare workers with distinct intensities of musculoskeletal symptoms. The analysis of such characteristics fosters discussions about the socio-productive impact of health workers’ morbidity and has the potential to provide relevant data to guide the adoption of management and public policy strategies.

## 2. Materials and Methods

### 2.1. Research Field and Participants

We investigated workers from 24 health services integrating the Brazilian Health System, 20 of which were Basic Health Units, in primary care, and the others were hospital and emergency care units. From four municipal hospitals, we interviewed workers from three hospitals and one emergency care unit as well. 

The sample is represented by workers who signed the formal acceptance term to participate in the research, including criteria of higher education and technical education. In Basic Health Units, we considered community health care agents and professionals who work directly with the user/patient (not including attendants and administrative assistants, among others). Other inclusion criteria were as follows: workers belong to the age group between 18 and 65 years, have at least 6 months of experience in their position and the company, have not been absent from work in the last 3 months, and work at least 20 h per week in this position. The exclusion criteria were (1) professionals in a pregnancy state; (2) amputees, wheelchair users, or workers with chronic mobility limitations; and (3) workers diagnosed with genetic diseases affecting mobility.

The participants were informed about the scope of the research, and after acceptance, they were instructed to sign the Informed Consent Form. This research followed national and international ethical guidelines and was previously submitted to the National Research Ethics Committee (CAAE: 79349617.6.0000.5188).

Using the list of health care units of the city of João Pessoa, Paraíba, Brazil, recruitment consisted of the probabilistic sampling of the analysis unit, characterizing it with stratified sampling. With the units of analysis already established, we developed a list of professionals working in the location. Next, we configured the arrangement of professionals that could compose the sample in each unit of analysis. Regarding the Basic Health Units, due to the restricted number of professionals, all professionals in the selected units who met the inclusion criteria were considered for the study. Whereas, in the specialized care units (hospitals and emergency care units), which have a higher number of professionals and sectors, the sector to be analyzed was initially selected by lottery, and then the same procedure was followed in the recruitment carried out in the Basic Health Units. 

We used Equation (1) to calculate the sample size of a finite population with methods that consider qualitative variables [21]. The sample size obtained was 323 individuals selected from an approximate population of 2000 workers.
n = (zg^2^·p·q·N)/(e^2^(N − 1) + z_g_^2^·p·q) (1)
where:

n = sample size;

zg = abscissa of the standard normal distribution fixing a confidence level g; 

p = estimation of the population proportion; 

q = 1 − p; 

N = population size; 

e = sampling error (maximum allowed difference between p and p).

### 2.2. Procedures

Data collection was carried out punctually and individually during the work shift in a restricted room with an appropriate environment for information collection. We recorded the frequency and intensity of musculoskeletal symptoms using the Nordic Musculoskeletal Symptom Questionnaire (NMQ), validated by Pinheiro et al. [22] as a morbidity measurement tool. The frequency and intensity of symptoms of pain, discomfort, and/or numbness in the previous 7 days were considered using a five-point Likert scale in each investigated region.

We evaluated performance using the Brazilian version of the Work Role Functioning Questionnaire (WRFQ), with construct validity tested comparing individuals with musculoskeletal disorders to healthy workers (α de Cronbach = 0.95; interclass correlation coefficient = 0.82 to 0.91) [23]. The questionnaire has 27 items, divided into five subscales (work planning demands, production demands; physical demands, mental demands, and social demands). And we use Teamwork Workload Scale (TWS) to assess the workload caused by team relationships. This questionnaire consists of 8 domains (team leadership, team orientation, performance monitoring, backup behavior, adaptability, trust, shared mental model, and communication) [24].

### 2.3. Statistical Analyses

The internal consistency of the questionnaire was evaluated through Cronbach’s alpha coefficient, considering the categories proposed by Landis and Koch [25]. The data obtained in the field were initially analyzed descriptively considering measures of central tendency and normality (Kolmogorov–Smirnov test).

To optimize the reliability of the outputs of the models presented below, we have scaled the target variables, this is because in models based on Euclidean distances introducing variables with different scales can increase the error of the estimates. For scaling the variables, the standardization technique was used [26]. After standardization, performance and team workload were dichotomized according to their positive (>0) or negative (<0) standardized values. This way, we have positive performance (containing standardized scores greater than zero) and negative performance (containing standardized scores less than zero) as variables. The same categorization procedure was performed on the team workload variable (positive = standardized scores greater than zero; negative = standardized scores less than zero).

Considering that the NMQ does not provide a score that merges the presence and intensity of symptoms, an Exploratory Factor Analysis (EFA) was performed in order to identify how the variables in this questionnaire are grouped and what score comes from each factor (for hypothesis 1). We assessed the EFA’s adherence to the data using Bartlett’s test of sphericity, the Kaiser–Meyer–Olklin (KMO) test, and the measure of sample adequacy (MAS) for each variable. The factors were extracted using the maximum likelihood method and the number of factors was determined based on the Kaiser and scree test criteria. The factor loadings were rotated using the Varimax criterion, from which the factor weights of each variable in each factor could be identified. We used the EFA scores to form homogeneous groups of workers so that we can then check the interaction between performance and team workload with each of the groups formed (for hypothesis 2). 

Group formation was performed using non-hierarchical cluster analysis, with data adequacy ensured through multicollinearity tests (variance inflation factor ≈ 1) [21]. We used the K-means algorithm as a measure of similarity between the scores obtained in the EFA to group the individuals according to their similarities. To form the groups, all the scores coming from the factors found were tested one by one as well as their combinations. The choice of the factor for the grouping was based on the identification of the one that generated the best group formation indicators, and the factorial scores resulting from it were tested for normality (Kolmogorov–Smirnov Test) and intergroup difference (Levene’s test, Kruskal–Wallis test, chi-square test, and Dunn’s hypothesis test). The choice of the number of groups was made based on test and retest, considering that, according to Hair et al. [27], “there is no standard, objective selection procedure”.

To analyze the relationships between the target variables, multiple correspondence analysis (MCA) was used (for hypothesis 3), relying on the standardized and dichotomized variables referring to performance and team workload, as well as the groups generated in the cluster analysis phase. In evaluating the model, the inertia and the percentage of variance for each dimension were considered. All the previously reported tests were performed in the R Core Team software (R Foundation: Vienna, Austria) [28], considering the significance level of 5%. The methodological flowchart of the research is presented in Figure 1.

## 3. Results

The Cronbach’s alpha coefficient was 0.92 (0.91–0.94; 95% CI), 0.83 (0.80–0.86; 95% CI), and 0.72 (0.67–0.77; 95% CI) for NMQ, WRFQ, and TWS, respectively, suggesting that these instruments showed acceptable internal consistency, indicating validity for the data collected via the questionnaire. As mentioned in Section 2.2, the NMQ has questions about the frequency and intensity of musculoskeletal symptoms in each body region. Therefore, if there is a certain frequency of symptoms, there is also an intensity attached to it, configuring it as potentially interrelated items, which justifies the high alpha value coefficient attached to this questionnaire [29]. It is understood that the frequency and intensity of these symptoms, although correlated, provide us with different information, making the high value (>0.90) of Cronbach’s alpha coefficient acceptable.

A total sample of 323 workers was obtained, with 50% of the sample consisting of individuals working in primary care, and the remaining individuals working in specialized care (hospitals and emergency care units). Among the questionnaires collected, 15 were excluded due to an extreme amount of lost data, making it impossible to form scores, which corresponds to only a 4% loss.

With 308 individuals and 18 variables, the data were considered eligible for AFE, with the adequacy of adherence confirmed by Bartlett’s test of sphericity (*p*-value = 0.000) and KMO equal to 0.79. MAS values for body regions ranged from 0.72 to 0.83, indicating the presence of a factor structure in the data with good indicators. Furthermore, 75% of the correlation coefficients between variables had values greater than 0.30. Therefore, we ensured the good suitability of the data for the AFE [27]. By the Kaiser criterion, the final model resulted in five factors selected to compose the analysis, with a cumulative variance of 0.68, which is above the value (>60%) suggested by the literature [27]. The scree test was an additional criterion used for choosing five factors.

In factor 1, symptoms are related to the region of the thighs; in factor 2, they are related to the region of the feet; and in factor 4, they are related to the region of the knees. In the case of factor 3, the frequency and intensity of symptoms in wrists, hands, and forearms were aggregated, close locations in which the occurrence and intensity of pain are associated with repetitive movements and constant tension. Finally, factor 5 aggregates the largest number of original variables and has a relevant characteristic, each variable of this factor concerns the frequency and intensity of regions that form the central axis of the body as follows: head, neck, upper back (thoracic region), and lower back. The factorial scores generated in this phase, for each sample case, and each factor, were used to group individuals by similar symptomatological characteristics. 

To form the groups, we adopted the factorial scores generated in factor 5. The tests with the other factors and combinations presented unsatisfactory indicators. Although factor 1 (frequency and intensity of symptoms in the thighs) has the highest eigenvalue in the analysis (8.021), it was not able to generate distinct groups, possibly because symptoms in this body area present themselves very uniformly in the sample concerning frequency and intensity.

The sample was allocated into three groups (G1 = 93, G2 = 102, and G3 = 110 individuals) with a sum of squares of 90.7%, denoting good model fit. The centroids were equal to 1.14, 0.12, and −1.10 for G1, G2, and G3, respectively. G1 includes the factorial scores of individuals who have a higher symptom burden than the mean (values greater than 1), G2 includes individuals who have a symptom burden approximately within the sample mean (values close to zero), and G3 included individuals who have a lower symptom load than the sample mean (negative values). The factor scores were statistically different for the three groups (Levene test: *p*-value = 0.011; Kruskal–Wallis test: *p*-value = 0.000), so we have internally homogeneous groups and heterogeneous groups. Figure 2 shows the 308 individuals considered in this sample and their position in relation to the generated groups related (different colors). The ordinate axis (y) identifies the factorial score generated by EFA.

The sociodemographic, lifestyle, and occupational characteristics of each group are shown in Table 1. The only differentiating factor between the groups was the sector in which the professional works with *p*-values prevalently greater than 0.05, one can understand that the distinction between the groups is at the symptomatological level and is not contaminated by sociodemographic or occupational characteristics. Regarding the work sector, after testing the groups pair by pair using Dunn’s hypothesis test, it was observed that G3 is different from the others. When compared with G1, it had a *p*-value = 0.008, and when compared with G2, it had a *p*-value = 0.019. Considering the percentage of professionals from specialized care in each group, one can then interpret that these professionals are significantly more present in the group where the musculoskeletal symptom load is lower, suggesting that they are less affected. The time of profession and work in the sector were statistically equal between the groups, and both had an average of more than 7 years in their respective positions, which shows that the workers’ answers were not influenced by their adaptation to the sector and work adjustment within the team. 

Although we have a high percentage of women in the overall sample (85.7%), our results are not exclusive to this gender, considering that we have 44 men (14.3%) included in the analysis. Furthermore, the high frequency of women in samples from healthcare professionals has been repeated in other publications [15,30], this is generally explained by the higher adherence of women to this type of profession. It is highlighted that men and women were equally distributed among the groups formed (Table 1, *p*-value = 0.128). 

By evaluating the correspondence between the variables through the ACM model, two dimensions were obtained, and the descriptive measures of this analysis are shown in Table 2. It can be seen that the two dimensions that describe the variability have approximately the same importance and, cumulatively, these dimensions have a total eigenvalue of 2.198 and inertia greater than 70% (0.733), denoting that they explain a large part of the data variability.

Figure 3 shows the proximity between the categories of variables in a two-dimensional space and should be analyzed considering their position as points in a plane and their horizontal and vertical distance, both with the same degree of importance. This means that the smaller the Euclidean distance between the points on the graph, the greater the association between these categories.

In this way, looking at the image, it is clear that the group of workers with the highest musculoskeletal symptom burden (G1) is closest to the category that denotes the greatest difficulties in team workload (positive teamwork). It can also be observed that “negative performance” is also very close to these two variables, especially to “positive teamwork”, forming a triad that exposes the structure of relationships between these variables. It should be clarified that in the tool used to measure team workload, higher values denote a greater workload imposed by team relationships, bringing a burden to the work. In moving away from G1, the relationships between performance and team workload moved closer together in two-dimensional space, as can be seen through the variables “negative teamwork” and “positive performance”.

Given these findings, it can be interpreted that in situations where the symptom load is reduced, the interaction between high performance and a favorable team workload is improved. Therefore, it is understood that the musculoskeletal symptom burden of workers can negatively interfere with the workload of the team and their performance. Finally, it is understood that the severity of the presentation of musculoskeletal symptoms has a potential effect on the overload of healthcare teams and can even narrow possible relationships between team workload and performance.

## 4. Discussion

Considering all the previously reported conjuncture, and considering the current literature, it is understood that organizations should facilitate a positive work environment in favor of a culture where the link between healthy professionals and productive organizations is encouraged. As far as health services are concerned, although training a professional has a positive effect on the quality of work, it alone is not enough to increase performance to adequate levels in unfavorable situations [31]. A team-based effort is probably the most flexible and available parameter for significantly improving the quality of care in health services [4].

Healthcare workers are a professional category substantially exposed to the most diverse sources of illness. They perform their responsibilities under pressure coming from various sources (users and management), many times in workspaces where the necessary tools to perform their responsibilities are deficient, where labor relations can be insecure, especially in situations where these relations are established by service contracts, as is the case of approximately half of the sample in this research (Table 1), where the organization of work can suffer interference from external factors (political factors, for example) and predispose to the formation of weak labor bonds.

All of these organizational characteristics bring suffering to the worker and implementation of a chronic illness process, which often manifests itself slowly by nonspecific symptoms, such as musculoskeletal symptoms, which, even for the worker, are often not characterized as a disease in itself or a disorder that needs resolution, but are treated as a normal result of the occupational routine. This is the naturalization of illness, which is strengthened by the social context of work and the economic and labor context in which the individual is inserted. 

As chronic health conditions worsen, with the onset of moderate and multi-site pain, an individual’s work experience is negatively affected [32]. In units with better ergonomic practices and where there is encouragement for a positive and supportive work environment that promotes employee involvement in decision making, trust, and cooperation, one finds workers with fewer work limitations [33,34].

In this regard, the role of healthcare managers is essential to detect early signs of dysfunction. Monitoring and management are vital to achieving efficient teamwork, with all the qualities needed for a safer healthcare system. Teamwork can protect the worker from exhaustion, depersonalization, and disengagement, common domains of burnout syndrome, and reduce the workload imposed by organizational demands [30,35].

The effectiveness of a healthcare team to achieve its goals can be improved if there is functional teamwork, in which the leadership and support role within an organization plays a prominent position. However, it is important to mention the contribution of structural work issues, which also affect the way teamwork is developed. To promote efficient teamwork, it is necessary that the minimum requirements for adequate patient care are met, such as equipment and availability of beds, among others [36], which is a major obstacle for the Brazilian health system, given the precariousness of the services.

An effective management approach would focus on making adjustments to work dynamics with the aim of reducing risk factors, as well as eliminating or minimizing factors that impact worker performance. Managing workers’ health conditions offers the potential for organizations to provide an efficient service, especially in the health sector where human resources play a central role, as well as reduce costs and increase productivity [37], with a strong propensity to reduce absenteeism and all its related effects [38].

In Brazilian legislation, there are a series of regulatory norms (RN) that establish criteria and limits for carrying out work under certain circumstances. The basic guidelines for health workers are established by RN-32, however, this standard cites only criteria related to biological risks. The RN-17 contemplates ergonomic criteria, however, this standard does not select an end activity, mentioning nonspecific thresholds. In one of its sections, RN-17 [39] advises that “The organization should implement an epidemiological surveillance program for early detection (…), analyzed and presented using statistical and epidemiological tools”. In this sense, it is possible to specify that in the case of healthcare professionals, a surveillance of musculoskeletal symptom load at the head and spine level is a priority, especially in healthcare professionals who provide basic care. 

Understanding that deterioration in workers’ health is more likely to occur in poorer psychosocial environments [38] and that team culture can be directly linked to the quality of work and safety in health care delivery [40] reinforces the importance of ensuring that work is planned and managed in a way that is beneficial to health. Workers do not need to be perfectly fit to perform their work responsibilities, however, the work environment and dynamics must ensure that the work is as suitable as possible to provide support to less able workers, ensuring that the work is beneficial and not detrimental to their health. 

This study has as its main theoretical implication the recognition that in health services the intensity of the worker’s musculoskeletal symptoms can interfere with the team’s work production. In services where professionals have high musculoskeletal symptom loads, there may be a gap in service production as we have a higher team workload and lower performance, therefore everyone, including the patient, may suffer from the effects. 

In practical terms, the main contribution is the problematization of the work configuration of a professional class that is significant in number and function. Guidelines for improving teamwork can be obtained, such as vigilance regarding the health professional’s symptom load, which can give clues as to how the team workload and individual professional performance behave. Professionals who voice complaints about musculoskeletal symptoms should be monitored for productivity because they are a group of professionals who more often have low performance and difficulties in team relationships. 

Such recommendations encourage a culture within the workforce that recognizes the impact of having workers who are suffering as a consequence of fatigue and physical attrition. From this, measures can then be taken to optimize teamwork, which will make the health services more efficient in view of the improvement of human capital, which can even bring benefits regarding the shortage of professionals in this area and the costs generated to the health system. As objective recommendations, we suggest: (1) keeping low frequency and intensity of symptoms in the spine and head region to preserve the performance and team workload; and (2) increasing surveillance in basic healthcare professionals. Future research should focus on the analysis of symptomatological conditions beyond the musculoskeletal ambit, and may include physical and mental compositions for a more amplified view of the problem, seeking to interpret team interactions to face complex situations. 

Among the strengths of the research is the use of a numerically significant sample obtained through the collection in multiple units of analysis, ensuring the possibility of generalizing the results to the enrolled population, and the use of robust statistical tools. Moreover, the combination of these robust statistical tools to address the research objectives adds a certain originality to the methodology. Separately, the application and results of the methods are well known in the literature, however, the authors have not found a similar methodology in the current literature in a combined manner. 

The main limitation of this study is the fact that many professionals work simultaneously in more than one healthcare institution, including in the private sector, which can contribute to the work overload of these professionals. Also, the use of questionnaires as the exclusive way to obtain data may have introduced some bias in the study due to the fact that this information may vary according to the expectations, previous experiences, or even personality traits of the interviewees. However, it should be noted that much of this data seeks to reveal how workers experience their work environment and how they react to it. These characteristics could hardly be captured in a mechanical way because, according to Cui et al. [10], other types of measurement may not be sensitive for measuring staff workload when the complexity of the scenario is an important influencing factor.

About the sample, our unit of analysis was the municipal public health network of João Pessoa/Paraíba/Brazil, that is, our sample is organized under the same management principles. Therefore, in this regard, workers have the same working conditions. However, each service has particularities, some services are only for cardiological assistance or pediatric assistance, requiring specific adjustments for each local need. We approached a probabilistically representative sample of the local population of health workers, and to minimize possible asymmetries in relation to specific working conditions, we had 50% of the sample obtained from basic care professionals and 50% from specialized care.

In addition, the sample’s professionals performed the same responsibilities, considering that they provide health care services; however, the tasks may vary within the same professionals in the same sector. For example, a physician on duty and a routine physician from the same unit have distinct tasks. The same tasks may be easily found in responsibilities in which there are low training requirements and high mechanizations, these less skilled workers are referred to by Grinza and Rycx [41] as blue-collar workers.

Therefore, the physical effort generated by these tasks may vary depending on the specific responsibilities performed. Table 1 shows that even though they are professionals with different demands (see professional) there is no statistically significant difference in the allocation of these professionals among the groups of high (G1), medium (G2), and low (G3) scores of musculoskeletal symptoms, that is, even with different levels of effort, they are proportionally allocated among groups. This can be explained by the fact that it is widely understood in the current literature that the effort generated in the execution is not the only factor generating musculoskeletal symptoms, and it is possible to cite factors such as mental demands of work, leadership profiles of the sector, job satisfaction, among others [13]. 

## 5. Conclusions

From the data previously presented, it is understood that higher performance is achieved when there is a favorable team workload, but this scenario is improved in situations in which the musculoskeletal symptom load for healthcare professionals is reduced. In a two-dimensional space, lower performance and worse team workload were closer to the group of professionals with high musculoskeletal symptom loads, which leads to the belief that symptoms can negatively interfere with both team relationships and team performance. These results suggest that to keep the performance of these professionals at favorable levels, it is also important to maintain a better team workload and a low burden of musculoskeletal symptoms.

## Figures and Tables

**Figure 1 ijerph-20-00742-f001:**
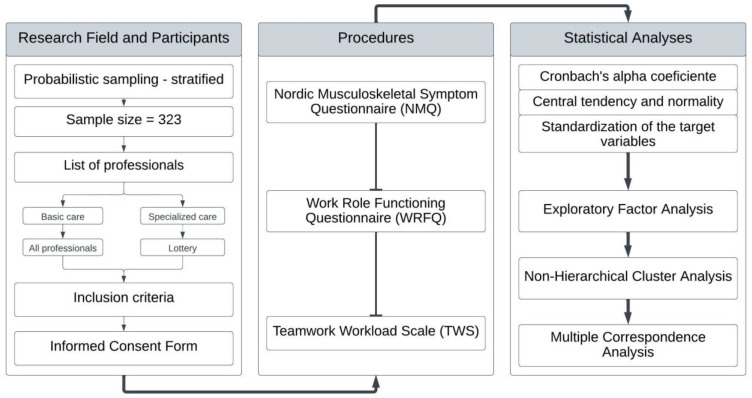
Workflow of the method.

**Figure 2 ijerph-20-00742-f002:**
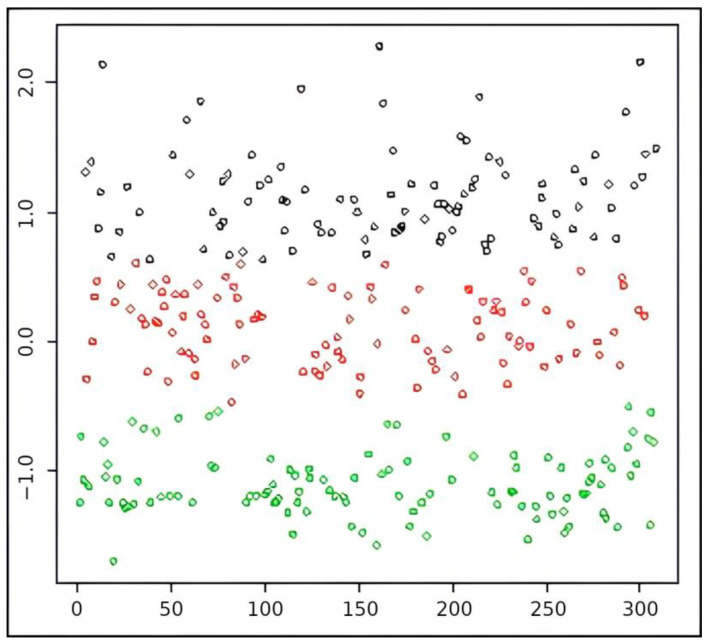
Sample grouping. Caption: G1 = black; G2 = red; G3 = green.

**Figure 3 ijerph-20-00742-f003:**
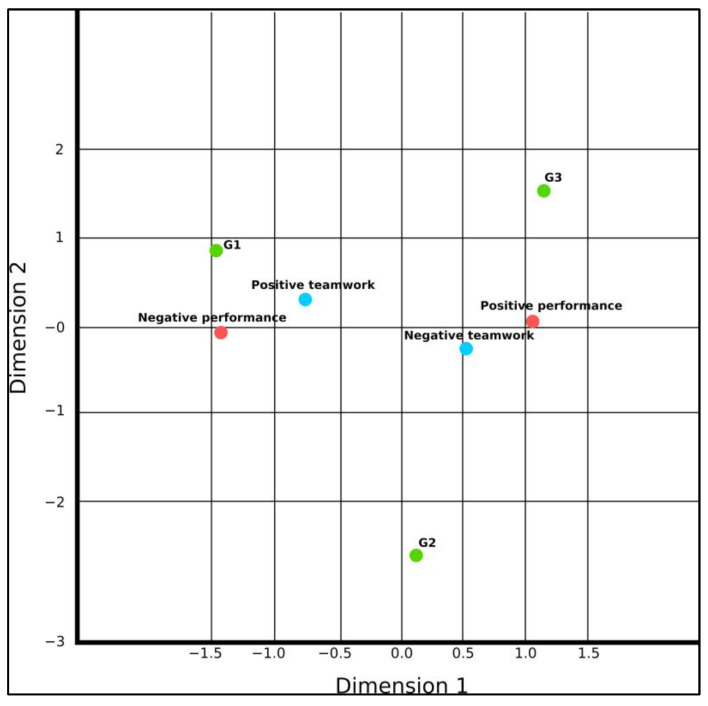
Correspondence between symptomatic groups, team workload, and performance.

**Table 1 ijerph-20-00742-t001:** Sample characteristics (mean ± standard deviation; percentage).

	G1 (n = 96)	G2 (n = 102)	G3 (n = 110)	*p*-Value
Sociodemographic and lifestyle				
Age	40.5 ± 11.1	40.9 ± 12.2	40.2 ± 10.7	0.996 *
BMI	26.4 ± 04.5	26.2 ± 04.5	26.3 ± 06.0	0.865 *
Female	91.70%	82.35%	83.63%	0.128 **
Married	48.95%	47.05%	49.09%	0.958 **
With children	65.62%	64.70%	60.00%	0.825 **
Smoker	04.17%	02.94%	04.54%	0.848 **
Does not drink alcohol	45.83%	55.88%	60.00%	0.228 **
Physical activity (weekly)	41.66%	50.98%	52.72%	0.337 **
Occupational				
Work sector				
Basic Health Units	59.38%	60.41%	48.95%	**0.033 ****
Specialized Care Units	40.63%	45.83%	65.62%	
Profession				
Physician	14.58%	09.80%	14.54%	
Nurse	21.88%	19.60%	19.09%	
Physiotherapist	07.29%	08.82%	06.36%	
Nurse technician	13.54%	16.66%	17.27%	
Speech therapists	02.08%	00.00%	00.90%	
Nutritionist	02.08%	02.94%	04.54%	0.176 **
Pharmaceutical	00.00%	02.94%	03.63%	
Occupational Therapist	00.00%	00.98%	00.00%	
Physical educator	00.00%	00.00%	02.72%	
Psychologist	01.04%	03.92%	03.63%	
Dentist	13.54%	06.86%	01.81%	
Social Work	04.16%	03.92%	09.09%	
CHA	15.63%	17.64%	11.81%	
Others	04.16%	05.88%	04.54%	
Educational level				
High school	06.30%	05.88%	03.63%	
Technical education	15.00%	18.62%	17.27%	
University graduate	20.00%	29.41%	36.36%	0.365 **
Specialization	56.00%	43.13%	39.09%	
Master’s degree	03.10%	01.96%	02.72%	
Doctorate degree	00.00%	00.00%	00.00%	
Post-doctoral	00.00%	00.98%	00.90%	
Working hours (weekly)				
20 h	06.25%	09.80%	07.20%	
24 h	02.08%	02.94%	04.54%	0.481 *
30 h	21.87%	21.56%	25.40%	
40 h	64.58%	57.84%	55.45%	
Other	05.20%	07.84%	07.27%	
Time of profession (years)	14.6 ± 10.4	14.0 ± 10.1	13.0 ± 09.5	0.688 *
Time working in the sector (years)	07.2 ± 05.9	08.1 ± 06.8	07.1 ± 06.7	0.427 *
Service provider	56.25%	49.01%	57.27%	0.405 **

Legend: BMI = Body Mass Index; CHA = Community Health Agent; * Kruskal–Wallis test; ** chi-square test; bold = significant *p*-values.

**Table 2 ijerph-20-00742-t002:** Descriptive measures of the Multiple Correspondence Analysis.

Dimension	Eigenvalue	Inertia	% of Variance
1	1.197	0.399	39.904
2	1.001	0.334	33.371
Total	2.198	0.733	-
Mean	1.099	0.366	36.637

## Data Availability

Not applicable.

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
