# Peer review of "Team Workload and Performance of Healthcare Workers with Musculoskeletal Symptoms"

_ijerph, 2022, doi:10.3390/ijerph20010742_

Round 1

Reviewer 1 Report

Thank you for allowing me to review the manuscript. This study is interesting, and I have a few comments to improve it:

1) Perhaps the authors could include a short 1-sentence background, and also the problem statement before they state the aim of study at the beginning of the abstract.

2) Lines 113-119, although there appears to be a reference as to where the sampling equation was extracted from, it would benefit more readers if the authors could explain what sampling equation is this (its name and description perhaps). 

3) Lines 181-184, according to past researchers, if the Cronbach alpha is too high it may suggest that some items are redundant as they are testing the same question but in a different guise (See https://doi.org/10.5116/ijme.4dfb.8dfd). Perhaps the authors could provide a justified reasoning (with citations) to safeguard themselves against this argument.

4) Lines 221 and 331, please avoid using words such as "you". Academic writing often requires us to avoid first-person point of view in favor of third-person point of view, which can be more objective and convincing.

5) It would be more scientific if the authors were to create hypotheses that linked up to the aim of study, and resolved via the statistical analyses.  

6) Please include limitations of study within the conclusion, and future directions of research. 

Thank you.

Author Response

1) Perhaps the authors could include a short 1-sentence background, and also the problem statement before they state the aim of study at the beginning of the abstract.

Answer: As suggested, we added a short introductory text to the beginning of the abstract (Lines 12-15).

2) Lines 113-119, although there appears to be a reference as to where the sampling equation was extracted from, it would benefit more readers if the authors could explain what sampling equation is this (its name and description perhaps). 

Answer: We added the details of the applicability and purpose of the equation to the paragraph (Lines 125-137)

3) Lines 181-184, according to past researchers, if the Cronbach alpha is too high it may suggest that some items are redundant as they are testing the same question but in a different guise (See https://doi.org/10.5116/ijme.4dfb.8dfd). Perhaps the authors could provide a justified reasoning (with citations) to safeguard themselves against this argument.

Answer: As suggested, we presented the clarification of this issue in lines 205-211. The authors are grateful for the suggestion.

4) Lines 221 and 331, please avoid using words such as "you". Academic writing often requires us to avoid first-person point of view in favor of third-person point of view, which can be more objective and convincing.

Answer: The snippets have been corrected (lines 250 and 383). The authors are grateful for the observation.

5) It would be more scientific if the authors were to create hypotheses that linked up to the aim of study, and resolved via the statistical analyses.  

Answer: The general hypothesis of the work was divided into hypothesis 1, hypothesis 2, and hypothesis 3, as presented in lines 71-80. We highlighted the statistical methodologies that contemplate each hypothesis in section 2.3, and each one of them refers to the analyzed hypothesis (See lines 173, 180 e 194)

6) Please include limitations of study within the conclusion, and future directions of research. 

Answer: We included the main limitations of the study in the last four paragraphs of the discussion section (lines (Lines 411-445).

Thank you.

Reviewer 2 Report

The paper is well written and organized. The research problem is interesting, and the results are good. However, the paper still needs to be further improved in quite a few points:

1. The method is well introduced. What are the novelties of the method? What steps does it have?

2. Figures of the method and experimental results are lacking. Figures of the workflow of the method or the architecture of the method is not provided. More visual results are needed to better present the empirical results.

3. Several health related works are missing, including but not limited to:

a. A Review on Recent Advances in Doppler Radar Sensors for Noncontact Healthcare Monitoring, IEEE Transactions on Microwave Theory and Techniques 2013

b. A spatial regulated patch-wise approach for cervical dysplasia diagnosis, AAAI 2021

Author Response

1. The method is well introduced. What are the novelties of the method? What steps does it have?

Answer: As suggested, we added a comment on the originality of the method to the discussion section (Lines 407-410).

2. Figures of the method and experimental results are lacking. Figures of the workflow of the method or the architecture of the method is not provided. More visual results are needed to better present the empirical results.

Answer: As suggested, we presented a flowchart with the methodological steps in Figure 1 (Page 5, line 200). In addition, we added the image of the position of each individual in the sample concerning the formed group (Figure 2, line 256).

3. Several health related works are missing, including but not limited to:

a. Review on Recent Advances in Doppler Radar Sensors for Noncontact Healthcare Monitoring, IEEE Transactions on Microwave Theory and Techniques 2013

b. A spatial regulated patch-wise approach for cervical dysplasia diagnosis, AAAI 2021

Answer: The cited works do not adhere to the theme of this article. Although they deal with issues related to human health, they do not refer to teamwork, performance at work, or musculoskeletal symptoms in the work context. In October of the current year, we carried out a new literary search to verify the existence of papers relevant to the theme. We included updates.

The authors are grateful for the suggestions.

Reviewer 3 Report

This paper offers important insights in relation to a particular category of professional work - health care. As in any other field, working conditions and tasks are critical to meeting the demands of the job while maintaining good personal health.

In addition, for each professional activity, there are established rules that must be respected because they ensure the safety conditions of the work and personal integrity. 

I kindly suggest clarifying the following details:

1) it is said  "our goal was to investigate the relationship between team workload and performance in healthcare workers with distinct intensities of musculoskeletal symptoms........and has the potential to provide relevant data to guide the adoption of management and public policy strategies." I suggest that you complete and report your research findings with specific legislation for this professional group: working conditions, level of exertion, level of education, work schedule/breaks, age, gender, etc.

2) it is said "workers from 24 health services integrating the Brazilian Health System, 20 of which were Basic Health Units, in primary care, and the others were hospital and emergency care units. .....The sample is represented by workers who signed the formal acceptance term to participate in the research, including criteria of higher education and technical education."

For the accuracy of the study, I consider it necessary that the people who agreed to participate in this study have the same working conditions and, above all, the same type of equipment (complexity, novelty, performances, etc.) to fulfil the work requirements.

I would also like to ask what level of education is required for this kind of professional occupation. Is it necessary to include persons in the sample who have a higher level of education? Do these persons perform a physical activity that requires effort?

The inclusions and extrusion criteria are very well described. 

3) For this study, the authors considered a population of 2000 workers. They have determined the size of the sample. I suggest first writing down the formula for calculating the sample size and then describing the meaning of the terms.

4) I noticed that there are only women in the sample. Are the conclusions and interpretation of the results valid for other genders as well, or not? I assume not. Based on the results, the authors can make some recommendations to improve working conditions and workload only for this type of gender. I suppose that the people in the sample are doing the same type of activities. 

Table..1... Check again whether the people who have a bachelor's or master's degree should be included in the sample. Do these individuals engage in physical activities because their job requires it?

5) Section 4. Discussions.... it is mentioned that "An effective management approach would focus on making adjustments to work dynamics with the aim of reducing risk factors, as well as eliminating or minimizing factors that impact worker performance. Managing workers' health conditions offers the potential for organizations to provide an efficient service, especially in the health sector where"... It is important to mention which are the relevant results compared to the specific legislation for this type of work because what has been mentioned above is general.

6) Based on their findings, the authors can make some recommendations regarding the intensity of the musculoskeletal load on the worker, i.e. the optimal level or degree of risk for a particular healthcare job. These findings can be used to update existing laws governing healthcare activity. 

Author Response

Question1) it is said  "our goal was to investigate the relationship between team workload and performance in healthcare workers with distinct intensities of musculoskeletal symptoms........and has the potential to provide relevant data to guide the adoption of management and public policy strategies." I suggest that you complete and report your research findings with specific legislation for this professional group: working conditions, level of exertion, level of education, work schedule/breaks, age, gender, etc

Question 5) Section 4. Discussions.... it is mentioned that "An effective management approach would focus on making adjustments to work dynamics with the aim of reducing risk factors, as well as eliminating or minimizing factors that impact worker performance. Managing workers' health conditions offers the potential for organizations to provide an efficient service, especially in the health sector where"... It is important to mention which are the relevant results compared to the specific legislation for this type of work because what has been mentioned above is general

Answer to suggestions 1 and 5: As suggested, we have added in Lines 363 – 372 a discussion of specific legislation and the contribution of this work.

Question 2) it is said "workers from 24 health services integrating the Brazilian Health System, 20 of which were Basic Health Units, in primary care, and the others were hospital and emergency care units. .....The sample is represented by workers who signed the formal acceptance term to participate in the research, including criteria of higher education and technical education."For the accuracy of the study, I consider it necessary that the people who agreed to participate in this study have the same working conditions and, above all, the same type of equipment (complexity, novelty, performances, etc.) to fulfil the work requirements.

Answer: We add some considerations about the research sample characteristics. See lines 422-445.

I would also like to ask what level of education is required for this kind of professional occupation. Is it necessary to include persons in the sample who have a higher level of education? 

Answer: The minimum level of education required for these professionals is technical education. A high level of education is not necessary, it is a professional option.

Do these persons perform a physical activity that requires effort?

Answer: We add some considerations about the research sample characteristics.  See lines 436-445.

Question 3) For this study, the authors considered a population of 2000 workers. They have determined the size of the sample. I suggest first writing down the formula for calculating the sample size and then describing the meaning of the terms.

Answer: As suggested, we changed the passage (lines 125-137)

Question 4) I noticed that there are only women in the sample. Are the conclusions and interpretation of the results valid for other genders as well, or not? I assume not. Based on the results, the authors can make some recommendations to improve working conditions and workload only for this type of gender.

Answer: Gender-related issues were described in lines 273-278.

I suppose that the people in the sample are doing the same type of activities. 

Answer: We add some considerations about the research sample characteristics. See lines 422-445.

Table..1... Check again whether the people who have a bachelor's or master's degree should be included in the sample.

Answer: Bachelor's and master's degrees should be included in the survey because the level of education refers only to training but does not refer to different activities.

Do these individuals engage in physical activities because their job requires it?

Answer: The question referring to physical activity refers to the lifestyle and not to the characteristics of the work. About the attributes of effort in the execution of the task, we added some comments in lines 436-445.

Question 6) Based on their findings, the authors can make some recommendations regarding the intensity of the musculoskeletal load on the worker, i.e. the optimal level or degree of risk for a particular healthcare job. These findings can be used to update existing laws governing healthcare activity.

Answer: We presented a recommendation regarding the intensity of symptoms on lines 397-400. It should be noted that the stipulation of a risk level is beyond the scope of this work.

The authors are grateful for the suggestions

Round 2

Reviewer 2 Report

My comments are addressed. I recommend to accept the paper.